# Creative Arts Therapy in the “Remote Therapeutic Response” Format in the Education System

**DOI:** 10.3390/children9040467

**Published:** 2022-03-26

**Authors:** Shir Korman-Hacohen, Dafna Regev, Efrat Roginsky

**Affiliations:** School of Creative Arts Therapies, University of Haifa, 199 Aba Khoushy Ave. Mount Carmel, Haifa 3498838, Israel; shirhacohen@gmail.com (S.K.-H.); roginskyefrat@gmail.com (E.R.)

**Keywords:** creative arts therapy, online psychotherapy, COVID-19, education system, remote therapeutic response

## Abstract

Many creative arts therapists work in the education system on a regular basis. As a result of the pandemic, all have had to treat students in a “remote therapeutic response” format. The aim of the present study was to map creative arts therapists’ perceptions of the “remote therapeutic response” in the education system. Semi-structured interviews were conducted with 15 creative arts therapists who participated in the study. The consensual qualitative research approach yielded seven domains: (1) the emotional experiences of transitioning to a remote therapeutic response; (2) the implementation of the remote therapeutic response; (3) benefits of remote creative arts therapy; (4) challenges in remote creative arts therapy; (5) remote contact with parents; (6) working in the educational system; (7) insights and recommendations. Although the findings show that creative arts therapists believe that remote creative arts therapy will never be a fully satisfactory replacement for most clients, remote work, despite its many difficulties and challenges, has also opened the door to new possibilities in the world of creative arts therapy in the education system.

## 1. Introduction

### 1.1. Remote Psychotherapy

Online psychotherapy has existed since 1961, when online group therapy was first proposed as a solution for clients living in isolated areas far from treatment centers [1]. Since then, online therapy has become more prevalent as a solution for clients who find it difficult to leave home because of illness, mobility issues, or as a way to continue treatment after moving, and has been found to be effective and beneficial [2]. Most studies on online psychotherapy have dealt with adult clients, but there are also a few findings on psychotherapy with children and adolescents. For example, Nelson and colleagues [3] compared CBT treatment to reduce symptoms of depression in children in a video versus a face-to-face format. The results, based on the assessment of 28 8- to 14-year-olds, indicated that video psychotherapy was effective and that changes occurred even faster than in face-to-face psychotherapy. There are also several case studies in the literature describing the online treatment of children and adolescents that were successful. For example, remote psychotherapy that included individual and family psychotherapy was provided to a child coping with depression [4], and there are reports of remote psychotherapy for children and adolescents with anxiety disorders [5].

### 1.2. Remote Creative Arts Therapy

The transition to remote creative arts therapy is particularly complicated, beyond issues related to the setting and therapeutic alliance, since it is based on the use of art materials, musical instruments, or theater props. This has prompted queries as to the suitability of creative arts therapy for online purposes, the nature of the triangular relationship (therapist, client, arts), and the effectiveness of the treatment [6,7].

Datlen and Pandolfi [8] described the transition of an open studio group for young adults with Intellectual Developmental Disabilities to the virtual space during periods of COVID social distancing. They created a WhatsApp group where the participants could share their artwork. The group was composed of five 17- to 23-year-olds. Most reported that the format was helpful as a substitute for face-to-face sessions. Shaw [9] described another format where an art therapy group took place via video calls. In this case, a group of three adolescents diagnosed with Anorexia received therapy based on free artmaking followed by verbal exchanges. The article discusses the art therapist’s feelings when the materials normally available to her clients could not be used [9].

A 2009 study described music therapy via Skype, which involved writing songs with an adolescent on the autism spectrum. Compared to face-to-face therapy, the adolescent was able to maintain eye contact for a longer period of time during remote therapy and demonstrated more creativity and self-confidence [10].

Kate Hudgins [11] described a successful form of remote psychodrama using the Therapeutic Trauma Spiral Model (TMS) and reported that many components of psychodrama are appropriate for online therapy, including the creativity and spontaneity that characterize psychodrama and the concept of surplus reality.

Shuper Engelhard and Furlager [7] presented case studies of remote dance and movement therapy with children. The difficulties in the online setting were related to the partial visibility of the body and the inability to provide the client with a sensory experience. However, the authors noted that the clients’ choices of which parts of the body to display on the screen informed the therapist as to their inner experiences. Another case study [12] described a number of group dance and movement sessions with adults coping with depression who were forced to switch to remote or open-space sessions because of COVID. The therapists listed the factors that contributed to the success of the treatment, including thinking outside the box, the clients’ previous familiarity with the therapist’s ways of working, the movements and the music, and the realization that remote sessions have become the norm.

### 1.3. Creative Arts Therapy in the “Remote Therapeutic Response” Format in the Education System

Creative arts therapies in the education system exist in many countries in different forms [13]. Recent studies in this field point to the contributions of the integration of creative arts therapies to the education system [14,15,16]. The COVID pandemic has led to sudden changes in the ways a therapeutic response can be provided by creative arts therapists in the education system when students cannot attend school or when there are interruptions in the therapeutic sequence as a result of lockdowns. This reality has forced therapists to shift, in conjunction with the entire education system, to distance therapy and adapt themselves to a new treatment format, which was rare before the pandemic. The present study was designed to map creative arts therapists’ perceptions of the “remote therapeutic response” in the education system. The term “remote therapeutic response” was used because in the early stages of the pandemic. It was not clear even to the therapists themselves whether the response format corresponded to real therapy or was merely a temporary therapeutic solution.

## 2. Materials and Methods

### 2.1. Participants

Fifteen creative arts therapists providing a “remote therapeutic response” in the education system took part. They ranged in age from 30 to 63 (M = 46.47) and had worked in the education system for 10 months to 25 years (M = 15.59). Eleven participants were also supervisors. The participants worked in various settings in the education system, from kindergarten to high school, in special education schools, and in regular schools where students with a variety of difficulties are integrated. Two worked in schools in the Arab society of the education system. The participants specialize in various creative arts therapy modalities (Figure 1). Only three had any previous experience with remote creative arts therapy or remote supervision. To respect the privacy of the participants, no table detailing their demographic characteristics is presented. In the acknowledgments section, we thank the participants who agreed to be mentioned by name.

Semi-structured interviews were conducted based on pre-prepared questions but were subject to change and open to expansion and adjustment according to the dynamics of the interview and the information that emerged [17]. The purpose of the interview was to obtain an initial assessment of the state of remote therapeutic responses in creative arts therapy in the education system during the first year of the COVID pandemic, its advantages and disadvantages, and the nature of the therapists’ remote relationship with the education system and parents. The interview was also designed to learn more about how the creative arts therapists worked, which approaches were successful and which were less so, their personal experiences when working remotely, and recommendations to enable therapists to adapt and feel more confident when engaged in remote arts therapy.

### 2.2. Procedure and Ethics

A letter of request to participate in the study was sent in October 2020 by one of the researchers to 28 creative arts therapists enrolled in advanced training in a course entitled “Remote Therapeutic Response in the Creative Arts Therapies”. Each therapist decided freely and anonymously whether to agree to be interviewed and if so, signed an informed consent form. The interviews, which lasted about an hour, took place via Zoom and were audio recorded solely for purposes of documenting the conversation. All information was collected anonymously. To do so, the participants were instructed not to mention any detail that could identify themselves or any other person. In cases where a person was inadvertently identified during the recording, the data were deleted from the recording immediately after the interview. The data were stored on protected platforms, and consent forms were kept separate from the transcripts of the interviews. This study was approved by the Ethics Committee of the Faculty of Welfare and Health at the University of Haifa (425/20) and by the Chief Scientist at the Ministry of Education (11451).

### 2.3. Data Processing

The data analysis was based on the principles of consensual qualitative research [17,18], which relies on phenomenological elements and aims to understand participants’ subjective experiences. Here, the process consisted of three stages. In the first stage, each researcher conducted a preliminary analysis of three interviews separately to identify and define the main domains that emerged from the data. Then the researchers met to reach an agreement on the key domains. In the second stage, all the interviews were coded according to these domains. In the third stage, the three researchers conducted a cross-sectional analysis to identify and define the core ideas, at which point all the data were reanalyzed according to these core ideas, with reference to their frequency.

In what follows, “most cases” describes a notion expressed in over 75% of all interviews, “some cases” refers to instances found in 25–75% of the interviews, and “a few cases” or “several cases” corresponds to fewer than 25% of the cases [17,18].

## 3. Results

### 3.1. The Emotional Experiences of Transitioning to a Remote Therapeutic Response

#### 3.1.1. The Creative Arts Therapists’ Experiences at the Beginning of the Pandemic

Eleven interviewees described feelings of confusion, shock, and stress during the transition to the remote therapeutic response at the beginning of the first lockdown. These feelings were possibly related to the rarity of a government injunction to limit people’s freedom of movement. Some linked their reactions to frequent fluctuations in the Ministry of Education’s guidelines for creative arts therapists, which could change overnight with no warning during the initial period: “There was so much pressure to adhere to the instructions… all of a sudden at ten o’clock at night you get new instructions for the next day, and you have to start from scratch in the morning”. Others attributed their panic to the need to learn new techniques quickly and the switch from their familiar creative arts therapy rooms to a computer format: “I did not know how to use Zoom or send a link, all these technical issues... it really stressed me out” (Table 1).

Five creative arts therapists discussed the chaos related to the state of emergency in Israel: “I felt it was more of an emergency situation, like when a war breaks out…. than remote therapeutic work”. These feelings prompted some therapists to engage in a rapid learning curve: “I turned the bedroom into an art therapy room during the day. It was a form of survival, it was either not working at all or making a change and that’s what I did”. Other therapists described their need to continue working and their sense of enthusiasm: “There was some joy at being able to respond, a lot of enthusiasm and strength”.

#### 3.1.2. The Adaptation Process

Many creative arts therapists realized that they needed to adapt to remote therapy because COVID was not going away any time soon. “Last year people were saying it would soon be over. This year people have realized that therapeutic thinking and planning have to be different.” The therapists described how in different ways, they quickly learned how to engage in remote therapeutic methods. Seven participants described how they searched for inspiration independently: “I’m a collector of any activity that goes online via Zoom”. Five therapists described learning from colleagues, for example, in supervision groups: “There were therapists in different modalities, each one contributed their own ideas, or their personal tricks of the trade, and games”. Some therapists described how they practiced the new activities, alone or with colleagues: “We practiced with each other. We started typing, testing the options, overcoming obstacles together”.

Eleven creative arts therapists said they gradually adjusted to the technology, which enabled them to feel more relaxed with respect to the remote therapeutic response in general: “If you had asked me half a year ago to deal with Zoom, it would have been beyond me. Then I learned how to operate it and it got easier”. Beyond the technological adaptation, five creative arts therapists also described their emotional process that included acknowledging the need to process events and find a breathing space: “I had to go through the initial shock... and afterwards, I felt I was really freer to pause and think about it”. The adjustment allowed some therapists to also enjoy learning a new skill: “It taught me new things that I find very interesting, I really enjoyed the success and being able to learn”.

#### 3.1.3. Continuing Instability

Two creative arts therapists expressed frustration and burnout as time went by, which was also affected by the frequent alternations between face-to-face and remote work: “The experience of working remotely was exhausting and frustrating and I really can’t do this anymore. It has lasted too long”.

### 3.2. The Implementation of the Remote Therapeutic Response

#### 3.2.1. Modes of Remote Therapeutic Response and Their Implementation

*Using the phone and WhatsApp.* Thirteen creative arts therapists reported using the phone or WhatsApp at various stages after switching to a remote therapeutic response or with certain clients. Six therapists said they began by using phone calls and texting at the start of the transition to remote therapy: “Before I started Zoom sessions, I did two weeks of WhatsApp conversations. I devised the infrastructure”. Five therapists noted the differences in therapeutic work between the first lockdown that began in the middle of 2019–2020 school year and therapies that began the next year (2020–2021): “Last year most treatments were over the phone because that’s what most kids had. They did not have access to a computer. This year it’s more often on Zoom, they know how to use it better”. Four therapists said that the therapeutic relationship was maintained even later by phone and WhatsApp when clients did not have access to computers at home: “In observant households some families have computer technology and others do not, so that the only thing left is phone calls”. Five therapists stated that in some cases, the decision to use phone calls or texting was related to the clients’ preferences or their ability to use Zoom: “One child I tried to treat through Zoom didn’t work because he was unable to sit down. So occasionally we talked on the phone or recorded messages”.

Six therapists stated that when they could not meet clients face-to-face for sessions or were hard to reach via Zoom, therapy was reduced to support: “Messages, phone calls, just to say that I’m available. That I understand, and are there to listen to them”. The same was true for preschoolers or clients with complex disabilities: “I would record myself reciting stories and songs, I would wear a costume and hold props up to the screen, then assign a task at the end of the activity, and they loved listening to it again and again”. Some therapists sent recordings, photos, and videos on WhatsApp even when Zoom sessions were taking place.

*Video call sessions.* All the creative arts therapists mentioned holding video call sessions via Zoom or WhatsApp. Fourteen therapists reported how they incorporated various arts and games: “We played on the screen, she had the game and I had the game, we showed each other the cards”. All the creative arts modalities were implemented. For example, in music therapy, a therapist noted: “With one child who had no musical instruments available, I asked him to take a pencil and we made rhythms”. Similarly, in dance and movement therapy a therapist commented: “ I asked them to dance with their parents, look at their bodies and communicate something”. In psychodrama, a therapist noted that: “I took dolls out of the regular therapy room and dressed them up, I told the student to do the same and I was able to get the dolls to have a conversation, her doll and my doll”. In art therapy: “I did the mirror exercise several times: I draw a line and they make a line”. The creative arts therapists described how they coped with the shortage of art materials in their clients’ homes: “We also did fine with pen and paper. The clients used what they had”. Two therapists described how they actually traveled to their students’ homes to deliver art materials to the door: “I brought them materials, I took a canvas and acrylic paints to a client”.

Eleven therapists also described capitalizing on the built-in features of Zoom or using other creative apps to incorporate arts and games into sessions. In particular, the therapists described the use of the screen and whiteboard sharing techniques for play or artmaking: “We tried to draw on the Zoom whiteboard. The client would select an image that interested her, we would put it on the screen and try to draw at the same time,” and some described incorporating interactive games. Three therapists said they combined the physical and digital formats: “I told the client to actually paint, not while in front of the computer, take a picture of it and send it to me and I would make a short film or sticker.” Three creative arts therapists nevertheless stated that they rarely used the applications in remote work: “I have not used the digital environment at all... I have never worked like this”.

The techniques implemented by many creative arts therapists also depended on the client’s environment. In some cases, the therapists asked them to use objects from home: “The children showed me objects and devices they had at home. We talked about them; ‘You have this kind of ball and you have this kind of other object’.” In other cases, the therapist was part of ongoing events at home during the session. Sometimes the therapists only interacted with the client by: “being with him in the room while he was playing. I watched and asked questions”. At others, the therapist was mixed into the entire household: “He really can’t sit still, so I got to know all the nooks and crannies of the house and I was even at the family dinner because he suddenly decided to sit down at the table”.

Many creative arts therapists also noted that verbal communication increased in remote therapies. Some described meaningful conversations and therapeutic work: “A client told me he could not make friends... it was a very exciting session, in one Zoom session we came a long way towards understanding this issue.” Some said that the exchanges remained limited to the game they were playing: “When the game isn’t there, he does not communicate, but when involved in the game, he communicates easily with me”.

*Limits of the remote response*. Despite the incorporation of the creative arts in the remote therapeutic response, 11 interviewees still felt frustrated by the narrow scope of artistic activities. The visual art therapists talked about how hard it was to engage in art materials with the client: “I haven’t found a good method for joint painting yet, or a way to do satisfying artwork together.” Others talked about how little material there was to work with, in particular in three dimensions such as clay: “The fact that it is impossible to work with plasticine and clay and three-dimensional objects is also difficult. It detracts from therapy.” Music therapists said it was complicated to play and sing together on screen: “Playing together or singing simultaneously really does not work out, it comes out a big mess, it is impossible to synchronize the rhythm.” In dance and movement therapy, the therapists struggled with the fact that only part of the body is visible on the screen: “Therapy when you only see part of people’s bodies... I feel that a lot of information I normally get is no longer there.” This was also true in psychodrama, where one therapist talked about having to overcome obstacles to work symbolically from a distance: “The discourse became much more verbal and challenging in the sense of symbolically expressing things”.

#### 3.2.2. The Therapeutic Act

*The essence of the relationship.* The sudden change in the format of the therapeutic response to students at the start of the pandemic raised a series of new questions as well as more general issues for the creative arts therapists. Many wondered whether the remote therapeutic response was indeed “therapy” or whether it was simply a way to maintain the relationship and support: “Do we still call it therapy, is it therapeutic support, to me they are different”.

*Treatment management.* Many creative arts therapists mentioned enacting major changes in treatment management. For example, one interviewee described the differences in her level of activity compared to a face-to-face session: “The variety in the art therapy room that you can choose from and access as you move around are not available remotely; I need to be more active.” Six therapists said that remote therapy needs more advance planning: “I see I need to sit and review my list of possible interventions, to see what will work and be interesting this time.” A few therapists reported that at times, despite their planning, the child brought something else to the online session: “(in a group) one child kept on talking excitedly, he showed all sorts of things and was so active that at some point I just dropped what I had planned and flowed with it.” Six therapists said that working with remote technology, which was new to both them and their clients, led to a more egalitarian relationship: “I felt we were trying to solve issues together—it would have been easier if I had known how to use the technology beforehand”.

*Therapeutic contract.* Some creative arts therapists mentioned issues related to the fundamentals of the therapeutic relationship, such as the therapeutic contract, coordination of expectations with clients and their parents, and the reflective discourse on the sessions: “Last year we had to digest all the changes and transitions. This year we put these contingencies into the contract.” The issue of coordinating expectations also arose with respect to the choice of format for the sessions: “We discuss it until we find what works for both of us so that the treatment will feel like treatment.” Some therapists noted that remote therapy also gives clients more control over when they start a session so that the responsibility for the relationship becomes more equally divided in some cases: “They were more responsible for their treatment because they could decide not to come (by not connecting to Zoom for example)”.

*Setting.* Many creative arts therapists noted that the boundaries of time changed during remote work, for example, in terms of the frequency of the sessions: “There was a daily conversation, sometimes only a few minutes, just to say hello.” The duration of the sessions was also affected by the transition to remote work: “We worked with them for shorter periods of time. If certain children did not want a session, the goal was just to keep in touch. A five-minute chat: ‘how are you’, and that’s it.” By contrast, three therapists said that the change in setting and treatment boundaries opened up new possibilities, for example, when working with adolescents: “They would send texts, and the fact that they could do so whenever they wanted and the therapist could respond at any time, was good for some clients”.

Another important component was finding a suitable environment for treatment in the students’ homes: “We needed to be much more careful about finding a quiet place, where you hear well, see well”. This was also true for the therapists’ homes: “I have a white wall behind me and bright light. But sometimes I have to find a quiet corner for myself. Be attentive to my setting so I will not be distracted.” Seven therapists noted that clients may not have any sufficiently private place in their homes for sessions: “The student does the session in the living room because she doesn’t have her own room.” Sometimes the therapist realized that there was another family member listening in on the conversation, which clearly interfered with the child’s privacy in therapy: “His mother is always there, you can hear her in the background, and I knew I could not ask all kinds of questions ...”. Ten therapists said they also had problems defining a private space in their own homes: “My son constantly interferes when I am on Zoom, no matter how much I explain to him that he shouldn’t.” Two therapists also raised the issue of uncontrolled exposure of the clients’ environment: “Suddenly I saw what her bedroom looked like, I saw family dynamics, things she did not want me to see.” However, many therapists noted that these glimpses of the clients’ environment also led to a better understanding of their worlds and relationships at home: “Seeing the child in his real environment and how the mother really behaves”.

*Initiating the therapeutic process.* Some creative arts therapists discussed the issue of starting a new treatment remotely: “It’s basically learning how to build trust and relationships on screen” compared to a therapeutic relationship that began in the arts therapy room and then switched to the digital space: “They felt we just translated it. It was not a new group, which is why it was less challenging”.

### 3.3. Benefits of Remote Creative Arts Therapy

#### 3.3.1. Benefits for the Client

Thirteen creative arts therapists noted that the remote therapeutic response allowed new events and insights to emerge within treatment: “I find that there is certain content that [emerges] precisely when the client is at home; it gives a sense of security.” Three therapists considered that this was particularly relevant to children on the autism spectrum: “Their experience is such that there is a screen between them and the other, which is transparent but still a screen. On Zoom they are in a safer place, and they reveal more of themselves.” Clients may also manifest abilities that do not appear in face-to-face sessions and are more willing to experiment; for example, in play and art: “Some children who refuse to draw on paper were willing to draw on the whiteboard where things can be erased... it is less committing.” In some cases, verbal communication was facilitated by the remote setting: “When we met face to face, it was less verbal. In the remote sessions, there was more discourse, it had more space.” This change at times persisted after the return to face-to-face sessions.

Some creative arts therapists noted that the remote therapeutic response helped maintain the continuity of treatment, by contrast to the fragmentation of the academic school year caused by lockdowns, illnesses, and quarantines: “I think the advantage is to keep in touch with the client, to convey a space of discourse, interest, caring, concern. Not to vanish.” Four therapists described how this later contributed to the continuation of face-to-face therapy: “It was very surprising and unexpected, something that leveraged the treatment when we returned to the therapy room”.

#### 3.3.2. Benefits for the Creative Arts Therapist

Some creative arts therapists indicated that working from home suited them at the time: “There was something in general about this period that suited me… this break in routine. It gave me some time to dwell on ideas.” Another interviewee said she felt freer when working remotely: “My creativity had more space to expand because I had more room to maneuver, I did what seemed right to me...”. Two therapists said that they felt that the new reality enabled them to develop professionally: “We had to acquire new tools. It forced us to think differently, to be mentally flexible.” This was also made possible by the larger number of advanced training courses: “I listened to a lot of lectures on Zoom during the lockdowns”.

### 3.4. Challenges in Remote Creative Arts Therapy

#### 3.4.1. Technical and Logistical Challenges

Most creative arts therapists reported challenges associated with technical problems on Zoom, such as being cut off, sound malfunctions, and internet crashes that affected the course of treatment: “The fact that there are ‘yes you heard me; no, you did not hear me’ malfunctions is an intrusion into the space of discourse.” Young children or clients with complex disabilities sometimes found it difficult to understand that they had to turn the camera toward their faces: “Some parents simply gave their phone to the child, so there were issues with muting or turning off the camera”.

Some interviewees reported that their home equipment was not adapted to Zoom: “You have to work with the computer you have at home, with your internet, with your chair. It is financially difficult anyway. It requires a lot of logistics, which is not very therapeutically oriented.” Many therapists also described technical issues associated with their clients’ lack of equipment: “Some people do not have a computer”, or in cases where the clients were not able to operate Zoom well: “The younger the children, the more they needed someone to connect to Zoom, and the times for the sessions needed to be set according to the availability of the people who could help them technically.” Sometimes these situations caused a real disconnect from the clients: “There was a student who did not have internet access anywhere in her neighborhood. I could not contact her, she was completely disconnected”.

Another logistical challenge for most interviewees was time management because they could theoretically hold treatment sessions throughout the entire day: “You can’t set a time, it’s just spread over the whole day and it’s awfully hard... I found myself on Zoom from morning to evening, in front of the computer, phones.” This problem was related to the expectation that therapists in the education system would make themselves more available to cope with the emergency situation, in particular at non-routine hours: “I make an appointment and the student tells me two minutes before ‘I can’t because my cousins just arrived, can it be postponed until...’ and I was very flexible so I let her postpone it to five o’clock... “ Four creative arts therapists described having to chase after the clients to have the sessions as scheduled. Eight therapists described time management as a major challenge, which also affected the treatment: “Hours that are not school hours, and sessions, and conversations with parents from morning to night... so there is really no let up from work.” Three interviewees also said that working on Zoom was physically tiring: “It’s terribly tiring. I try not to look away [from the screen] because it might be interpreted as abandonment by the client. Fatigue and great commitment”.

#### 3.4.2. Resistance during Remote Creative Arts Therapy

Many interviewees reported clients who were not interested in remote contact at all, which created a disconnect during the lockdowns and quarantines: “One child was not even willing to talk to me on the phone when he was in quarantine. It was a complete disconnect.” Several therapists noted that remote creative arts therapy had become just another Zoom session in a full day of classes and had lost its special qualities: “Some students refuse to connect, and you do not know if it is because of Zoom or not… The therapy session became like another class, as if I were a teacher.” Twelve creative arts therapists said that some clients, especially adolescents, refused to turn on their cameras or orient it at their faces: “(The client) was embarrassed about talking to me on the computer, so he turned the camera to the side, was not in the frame and then talked to me. It’s hard, not seeing eyes, or the face.” Five therapists also talked about distractions on the computer: “I realized that he was only half with me and that he was playing a game at the same time. We were able to maintain the connection but it was not like in the room”.

#### 3.4.3. Lack of Close Contact and Body Language

Fourteen creative arts therapists described the lack of close contact or observation of body language as major challenges in remote therapy: “There is a great lack of a sense of closeness, or access to non-verbal communication and body language. Often, I can sense clients even without them talking, but on Zoom I just can’t.” Another therapist talked about starting therapy on Zoom: “My first session with some students was on the screen, and I found it difficult to sense them”. Two interviewees also commented that the lack of physical presence made it difficult to remain silent during a session: “It’s not like in a face-to-face session when a child chooses to be quiet, where I allow that quiet moment to unfold by just being there. On Zoom you can easily be disconnected”. This issue was particularly pronounced with clients who found it difficult to express themselves verbally: “A student who sometimes finds it difficult to say hello can turn her head towards me to signal ‘Look, I’m here’. For low functioning children or those who do not speak Hebrew fluently, Zoom is problematic: “I have one client who speaks Hebrew but I’m not entirely sure he understands everything. I feel like I can’t get through to him on Zoom and reach the painful places”.

#### 3.4.4. Challenges in Maintaining Group Creative Arts Therapy

Some therapists noted that in remote group creative arts therapy, clients can interfere with the session: “There was one client who really ruined it for everyone else. This can happen in class, only here you can’t send [the student] out to chill, get a drink of water and come back.” In addition, some children found it difficult to stay attentive during remote group creative arts therapy: “In the group I felt they were not there, not involved, not attentive to each other”

#### 3.4.5. The Challenge of Being a Creative Arts Therapist during This Period

Some interviewees felt deprived of their skills and toolbox as a result of the transition to remote sessions: “Suddenly all your tools, your language, your modality [are gone]. Like someone pulled the rug out from under your feet.” Many therapists had to struggle to maintain sessions: “How do I keep on doing this for one whole hour? After all, in therapy the client is told that s/he can’t leave, and that if s/he is bored, s/he goes on being bored. On Zoom I can’t say that.” Some of the interviewees felt diminished: “I felt that my creativity and spontaneity had declined, even my listening was not the same.” Two therapists commented on the issue of creating a mental separation between home and therapy: “In my art therapy room I am more of a tabula rasa than at home”.

### 3.5. Remote Contact with Parents

#### 3.5.1. Closer Connections than Usual

Ten creative arts therapists commented that during the COVID period, there were more exchanges with clients’ parents compared to routine work: “Some parents who were never in touch suddenly asked about the child and talked about themselves every week.” One interviewee said that the parents, in fact, replaced the teacher as the link between the creative arts therapist and the student: “The Ministry of Education did not encourage much contact with parents in the past; we tended to be in closer touch with the educational staff. Suddenly parents have become the facilitators for their children, they are more involved.” Many therapists noted that parents were more available as a result of the changes in session format and flexibility in hours: “I was very happy to see that in the intakes it was really easy to set a time, because suddenly it was possible in the evening, suddenly both parents could attend.” Eleven therapists said that they attempted to support parents during this period: “I was in close contact with all the parents of the younger clients, including giving parental guidance on Zoom”.

#### 3.5.2. Parental Contributions to Managing the Remote Therapeutic Relationship

Seven interviewees stated that communicating more frequently with parents helped them understand the background to their children’s behavior: “When I could not always reach the client, at least I heard from the parents how he was.” Having the sessions in the home environment also made it possible to involve the parents in therapy: “She asked to include her father in a session because it was very difficult for them... this was made possible by Zoom and by being at home.” However, six therapists noted that parents sometimes intervened without coordination or invitation: “She intervened in the session when it was not needed. She did so out of interest but I felt it was a bit of an invasion of the child’s privacy as well”.

The novelty of the situation and the more intense communication with parents made many interviewees realize the importance of coordinating parental expectations and establishing a type of contract with respect to privacy during sessions, meeting times, help organizing materials, and other issues: “They are the only ones who can give their child privacy. They are the ones to allow the child to have art materials in the room. It requires cooperation.” This coordination with parents was even more important with very young children or clients with complex disabilities: “The parents of kindergarten children, we depended on them to start the Zoom session… The parents were in the background and helped technically and to focus the child.” A number of interviewees also said that the parents helped get the child ready for the session: “His mother sent me a text that she told him he would make music with me and he said that he was interested.” Compliance from adolescents was more complex. The nature of the relationship between clients and their parents affected their ability to help them: “It’s no longer a small child having problems that parents are responsible for solving... There are all sorts of other things related to the child-parent relationship”.

#### 3.5.3. When Parents Are Not Available

Some creative arts therapists described parents who were completely uninterested in contacting them during this period: “Some parents told me ‘talk to the child’, like, what do you want from me,” or were very busy handling the needs of several children. Sometimes this attitude affected the treatment sequence: “A student I worked with... nothing can be scheduled with her mother. I was able to work with her once or twice... I could not maintain a sequence”.

### 3.6. Working in the Educational System

#### 3.6.1. Contact with the Educational Staff

Many creative arts therapists said that their relationship with the school and the staff became stronger as a result of remote accessibility and the sense of urgency. This closer connection made joint thinking and teamwork possible but also provided a time to vent: “We were in touch, all the time. Any time there was a problem, the teachers or the principal would turn to me and we thought about what to do, the division of roles.” A few therapists noted that the special education teams were in closer contact than teachers in regular education and that the relationship was less reciprocal: “In regular education it is different, the teachers have enough on their minds. They do not always understand the nature of creative arts therapy.” In another aspect, many therapists noted that working remotely made it much simpler to hold staff meetings, and even attend meetings that they usually missed because of technical constraints or work in several frameworks: “If there is a kindergarten staff meeting, I take part on Zoom if it is not my day at the kindergarten, instead of running from place to place”.

#### 3.6.2. Contact with Officials in the Ministry of Education

Many interviewees noted the importance of their interactions with their supervisors at the Ministry of Education who provided technical and emotional support: “It was really powerful. I learned a lot on Zoom from my supervisors. They were really there for us and supported us during supervision, through courses, there was really a lot of moral support and reassurance”. The supervisors agreed: “So far, they have given me a free hand to supervise. It has been an amazing containment because we meet every week, work with art materials and develop methods.” However, two creative arts therapists felt that the supervision was not sufficient: “There were sessions with my supervisor but it did not meet my needs. I think that holding, supporting and accepting what was happening were missing”.

Two creative arts therapists and supervisors noted issues related to the bureaucracy of the Ministry of Education as a supervisory body, which intensified during this period: “There is something called a follow-up form. During the previous lockdown we were home, but once it expanded to work at school as well, I felt like it was ‘Big Brother,’ it’s very burdensome, it generates a sense of mistrust.” According to a number of interviewees, some clients had to cancel in order to adjust their schedules to pandemic regulations. This meant that, at times, it was impossible to meet work requirements for hours of therapy: “You are committed to working and you understand what they are going through, and you do not want to increase the stress and burden… So sometimes instead of working six hours you work less”.

### 3.7. Insights and Recommendations

#### 3.7.1. Internal Strengths and Resources Contributing to the Work of Creative Arts Therapists during This Period

Most creative arts therapists indicated that flexibility enabled them to make the sudden change to remote therapy: “I understood that there was no other solution, the world has changed and I had to adapt. This made me very intent on my work.” Flexibility was also expressed in relation to the setting: “I realized that there are a multitude of possibilities. Therapy is not restricted to a permanent and very rigid setting as Freud or others stipulated, we are in another era where the outside goes inside and vice-versa.” Other factors mentioned were creativity, optimism, and the ability to learn from experience: “The therapists, thanks to their creativity and playfulness were able to take advantage of the situation and insist on treatment and it won over the hearts of parents and children.” Three interviewees said that it was very important to maintain boundaries during this period and be able to listen to oneself in order to be able to work remotely: “Mostly keeping things tidy, that’s also boundaries. That’s the thing that saved me in all the chaos. I’m flexible but I tried to keep my working hours”.

#### 3.7.2. Feelings of Self-Worth and Professionalism

Six interviewees said that the success of remote creative arts therapy strengthened their sense of confidence as therapists: “I learned to trust myself. I felt valuable, that I was able to conduct most treatments that way, that I knew what I was doing.” Two therapists also described the ways this was reflected outwardly, in terms of feeling more appreciated by the staff: “Awareness of the profession has grown. There are more demands, from the kindergarten teacher, from the occupational therapist, from the speech therapist for our presence as creative arts therapists”.

#### 3.7.3. Recommendations

Three creative arts therapists referred to the need to adopt new perspectives and to learn from other professions to shift optimally to a remote setting: “I sometimes feel the need to get out of the box, and look at other fields.” Two interviewees suggested incorporating technology in the future in face-to-face creative arts therapy: “I have no doubt that we will take some of the technological tools back to the arts therapy room.” Twelve interviewees stated that they would prefer to work face-to-face in the future: “My initial preference is always face-to-face,” but also said that remote creative arts therapy would be useful in some cases, for example, when a client does not come to school occasionally: “It’s worth thinking about a sick child... there are children who have school anxiety. It also makes it possible to reach out to them,” or for health reasons: “People in hospitals who are really isolated, for example children with cancer who do not have an immune system.” Two interviewees also commented on the possibility of continuing sessions during vacations, when schools are closed: “Working in the summer with special education children, which is exactly the time when no one is watching over them and they are at risk”.

## 4. Discussion

This study was designed to map creative arts therapists’ perceptions of the “remote therapeutic response” in creative arts therapy in the education system. The findings indicate that the onset of the pandemic was characterized by complex emotions, alongside a massive mobilization on their part to find creative ways to continue therapy. The interviewees mentioned that independent learning, as well as help and consultation with colleagues and supervisors, contributed to their adjustment process. Recent studies have shown that supportive relationships with colleagues can enable therapists to better make the transition to online work [19]. The interviewees’ investment enabled therapy to take place, but their enormous dedication along with the ongoing instability that characterized the period sometimes led to feelings of burnout. This sense of burnout was acerbated by insufficient technological resources. The interviewees reported that they had to use their home equipment, which was not always up to date or adapted to Zoom. This suggests that a dedicated budget should be devoted to providing creative arts therapists with the appropriate equipment. In addition, the interviewees stated that their busy agenda and the intense intersection of work and home was extremely problematic. Clearer definitions of work hours and agendas should be considered in future guidelines.

The findings also showed that the transition to a remote therapeutic response restricted the creative arts therapists’ use of the arts, which sometimes undermined their confidence as therapists. Similar feelings have been described in other reports and case studies. For example, Shaw [9] described her feelings as an art therapist during the transition of a group of adolescents with Anorexia to remote art therapy. She felt it was harder for her to produce a containing feeling when she lacked the traditional tools of art therapy. However, the present study also pointed to a variety of ways in which therapists integrated the arts in all modalities. The literature also shows that the arts can be successfully integrated into remote therapy with appropriate adjustments. For example, a survey of music therapists found that pre-recording music was useful to the therapists in the sessions [20]. The therapists also reported a number of important components that helped them establish a therapeutic relationship remotely, including greater motivation on their part, a clear contract with respect to the setting, and defining appropriate boundaries in terms of time, the nature of the relationship, and place. Case reports and other studies have underscored the importance of the setting in remote therapies. For example, a survey of art therapists in the U.K. found that adjusting the therapeutic contract to remote therapy and clearly defining it around issues of time and location of therapy, the format, and communication between sessions, were particularly important and meaningful [19].

The findings also pointed to new opportunities that emerged in remote therapy, such as promoting greater confidence and self-expression in some clients, for example, for clients on the autism spectrum. This finding also emerged in another study that examined music therapy for a boy on the autism spectrum, which found that in remote therapy, compared to face-to-face therapy, the boy maintained more eye contact and expressed more self-confidence and creativity [10]. In the context of the benefits inherent to remote therapy, a recent preliminary study found that clients perceived their therapists as more empathetic and supportive than in standard therapy [21]. These preliminary findings hint there may be additional benefits of this mode of treatment.

The remote setting also changed the therapists’ relationship with their clients’ parents. Previous studies on the integration of creative arts therapy in educational settings have found that contact with clients’ parents is often unsatisfactory given their low level of commitment or availability coupled with the dearth of resources allocated for this purpose, such as dedicated therapists’ working hours. Some therapists only see parents at the beginning and end of the school year, and these parents do not play a significant role in the therapy process [22]. However, the relationship with parents took a significant turn during the first year of the COVID pandemic, when parents were responsible for enabling the therapy to take place. The video calls provided new possibilities for contact, and the therapists were willing to engage in continuous and supportive exchanges. Hence remote contact with parents could be considered as a solution in the future but should be budgeted and regulated. In systemic work, alternatives were devised to allow therapists who work in multiple establishments to take part in staff meetings remotely. Proper budgeting of remote call hours would allow therapists to attend staff meetings on a routine basis.

This study reinforces previous findings showing the significance of online therapy in instances where face-to-face therapy is not possible and its effectiveness in crisis situations [2,23]. The changing situations make it clear that the online format can be an advantage in cases of inaccessibility, illness, and emergencies. It would therefore be worthwhile to define a range of situations in which the use of a remote therapeutic response is authorized and budgeted.

### Limitations and Suggestions for Further Research

This preliminary study only examined a specific time period from the beginning of the pandemic and the transition to remote creative arts therapies up to the end of the second lockdown in October 2020 in Israel. Since then, changes have taken place in the field in terms of medical breakthroughs and in society, such that further research should examine how they have influenced remote therapy. This study also provided a broad overview of creative arts therapists’ perspectives who all work in educational settings, although the populations in question are very diverse. Further research should focus more deeply on specific populations to better understand their characteristics and needs. Finally, the present study examined the perspective of creative arts therapists. Further conclusions about the significance of therapy and other factors that can promote it could be drawn from a study examining the experiences of clients and their parents.

## 5. Conclusions

These findings raise numerous questions about the nature of remote creative arts therapy in general and how it is conducted within the education system in particular. These questions touch on the shortcomings experienced by the creative arts therapists during the COVID period but also the innovations. The setting, degree of exposure, and extent of privacy underwent dramatic changes in the period described in the study and affected the nature of therapy in different ways. Sometimes they caused embarrassment, problems of intimacy, and fear of overexposure, while in others, they actually allowed a closer connection and observation of facets of the lives of clients that would not otherwise be revealed, which provided the therapists a chance to intervene in a new way. Recent articles have begun to discuss the ethical implications of using the Internet for privacy and confidentiality [24]. Future work should continue to explore this in various forums in the therapeutic world, including in training programs for new therapists. In some cases, ecological therapeutic work was made possible by the creative arts therapists in extensive collaboration with the educational staff and the clients’ parents.

Although the findings suggested that the creative arts therapists believed that remote creative arts therapy will never be a fully satisfactory replacement for most clients, remote work, along with the many difficulties and challenges, has also opened the door to new possibilities in the world of creative arts therapy in the education system. The question as to the nature of the therapeutic response, whether it is therapeutic support or real therapy, continues to preoccupy the therapeutic field. What initially appeared to be an unsatisfactory temporary response was replaced by a growing understanding of the therapeutic implications inherent to it. Along with the long-awaited return to routine, it seems inadvisable to completely close this door without examining what is worth preserving.

## Figures and Tables

**Figure 1 children-09-00467-f001:**
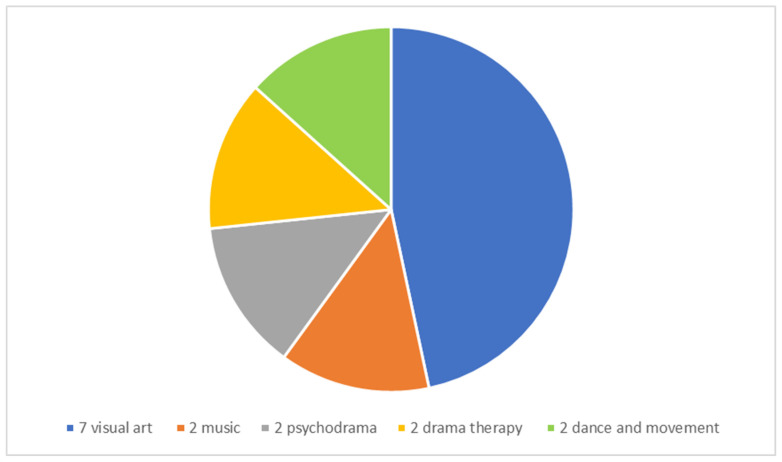
Instruments.

**Table 1 children-09-00467-t001:** Creative arts therapy in the “Remote Therapeutic Response” format as reported by participants: Domains and core ideas.

Therapists’ Emotional Experiences	Implementation of Remote Therapeutic Response	Benefits of Remote Creative Arts Therapy	Challenges in Remote Creative Arts Therapy	Remote Contact with Parents	Working in the Educational System	Insights and Recommendations
Beginning of the pandemic The adaptation process Continuing instability	Modes of remote therapeutic response Phone and WhatsApp Video call sessions Limits of the remote response The therapeutic act Essence of the relationship Treatment management Therapeutic contract Setting Initiating the therapeutic process	Benefits for the client Benefits for the creative arts therapist	Technical and logistical challenges Resistance Lack of close contact and less body language Maintaining group therapy Being a creative arts therapist during this period	Closer connections than usual Parental contribution to managing therapy When parents are not available	Contact with the educational staff Contact with officials in the Ministry of Education	Internal strengths and resources contributing to the therapists Self-worth and professionalism Recommendations

## Data Availability

The data are not publicly available due to ethical restrictions.

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
