# Peer review of "Creative Arts Therapy in the “Remote Therapeutic Response” Format in the Education System"

_children, 2022, doi:10.3390/children9040467_

Round 1
Reviewer 1 Report
I believe that the present critical issues arise:
- in section 2.1 line 80
start a series of numerical lists, it would be better if these lists were inserted in a numerical version, perhaps using a table or an infographic, it would make the reading more fluent and allow you to observe the creative process in its entirety.
- in the results section: 3.1.1 line 143
there are references to the online mode and the perception of emergency, it would be interesting to describe this area better
- in section 3.2.1 line 177
it would be interesting to summarize these results through a summary table, it would make the reading more fluent.
- in section 3.2.1.2 line 224
it would be important to create a "limits of this instrument" section
- line 253 this article could be cited, it sees me involved, so I would not want it to generate conflicts of interest, I link it anyway: https://doi.org/10.3389/fpsyg.2021.671790
- in section 3.2.2.1 line 260
the authors distinguish between therapy and therapeutic support,
it would be interesting to implement this part, I think it would make the article very original and would set the stage for a conversation in the scientific field of quality
- in section 3.4.1 line 373
there are studies that can be cited on the "false availability" effect
https://doi.org/10.1007/s11920-015-0629-2
https://doi.org/10.1176/appi.ap.35.3.168
Author Response
We would like to thank both reviewers for their important suggestions.
The whole paper was re-edited.
Reviewer 1
- in section 2.1 line 80
start a series of numerical lists, it would be better if these lists were inserted in a numerical version, perhaps using a table or an infographic, it would make the reading more fluent and allow you to observe the creative process in its entirety.
We have added Figure 1. We hope this makes it easier to read.
- in the results section: 3.1.1 line 143
there are references to the online mode and the perception of emergency, it would be interesting to describe this area better
We added a sentence. We hope it is clearer now.
- in section 3.2.1 line 177
it would be interesting to summarize these results through a summary table, it would make the reading more fluent.
We added table 1.
- in section 3.2.1.2 line 224
it would be important to create a "limits of this instrument" section
We added "Limits of the remote response" section (3.2.1.3).
- line 253 this article could be cited, it sees me involved, so I would not want it to generate conflicts of interest, I link it anyway: https://doi.org/10.3389/fpsyg.2021.671790
We added this reference to the Discussion section.
- in section 3.2.2.1 line 260
the authors distinguish between therapy and therapeutic support,
it would be interesting to implement this part, I think it would make the article very original and would set the stage for a conversation in the scientific field of quality
Please see the end of the Literature Review section and also in the Conclusion section.
- in section 3.4.1 line 373
there are studies that can be cited on the "false availability" effect
https://doi.org/10.1007/s11920-015-0629-2
https://doi.org/10.1176/appi.ap.35.3.168
We added one to the Conclusion section.
Reviewer 2 Report
Art Therapy is based on visual arts only, Expressive Therapy or Creative Arts Therapy is all modalities (art, music, dance, etc). In 1.2 section, the author has the heading Remote "Arts Therapy," the correct heading should be Remote Expressive Therapy or Remote Creative Arts Therapy. The participants in this study were Expressive Therapists or Creative Arts Therapists, not Art Therapists. Advise replacing Arts Therapy with Expressive Therapy or Creative Arts Therapy for an accurate description of the research study.
The display of information [In what follows, the phrase "most cases" describes a notion expressed in over 75% of 128 all interviews, "some cases" refers to instances found in 25-75% of the interviews, and "a 129 few cases" or "several cases" corresponds to less than 25% of the cases] seems haphazard. Why not provide the exact percentages, for example there is a hugh difference between the range of "25-75 % of the interviews." There are only 15 participants. The use of percentages presented here is deceptive. To make sense of the data the information provided should be accurate. For example, 14 out of 15 participants reported... One participant commented that... Three participants found that....
The authors need to be more specific in presenting the results of their study. Overall, the results are interesting and will add to the body of research on Creative Arts Therapy.
Author Response
We would like to thank both reviewers for their important suggestions.
The whole paper was re-edited.
Reviewer 2
Art Therapy is based on visual arts only, Expressive Therapy or Creative Arts Therapy is all modalities (art, music, dance, etc). In 1.2 section, the author has the heading Remote "Arts Therapy," the correct heading should be Remote Expressive Therapy or Remote Creative Arts Therapy. The participants in this study were Expressive Therapists or Creative Arts Therapists, not Art Therapists. Advise replacing Arts Therapy with Expressive Therapy or Creative Arts Therapy for an accurate description of the research study.
We changed it to Creative arts therapy.
The display of information [In what follows, the phrase "most cases" describes a notion expressed in over 75% of 128 all interviews, "some cases" refers to instances found in 25-75% of the interviews, and "a 129 few cases" or "several cases" corresponds to less than 25% of the cases] seems haphazard. Why not provide the exact percentages, for example there is a hugh difference between the range of "25-75 % of the interviews." There are only 15 participants. The use of percentages presented here is deceptive. To make sense of the data the information provided should be accurate. For example, 14 out of 15 participants reported... One participant commented that... Three participants found that....
The division was made according to Clara Hill's recommendations (see sources in the text). However, some places we indicate exact numbers to make the count accurate.
The authors need to be more specific in presenting the results of their study. Overall, the results are interesting and will add to the body of research on Creative Arts Therapy.